# NRF2 in Viral Infection

**DOI:** 10.3390/antiox10091491

**Published:** 2021-09-18

**Authors:** Angela Herengt, Jacob Thyrsted, Christian K. Holm

**Affiliations:** Aarhus Research Center for Innate Immunology, Department of Biomedicine, Aarhus University, 8000 Aarhus, Denmark; angelaherengt@biomed.au.dk (A.H.); thyrsted@biomed.au.dk (J.T.)

**Keywords:** NRF2, virus, immunology

## Abstract

The transcription factor NRF2 is central to redox homeostasis in animal cells and is a well-known driver of chemoresistance in many types of cancer. Recently, new roles have been ascribed to NRF2 which include regulation of antiviral interferon responses and inflammation. In addition, NRF2 is emerging as an important factor in antiviral immunity through interferon-independent mechanisms. In the review, we give an overview of the scientific progress on the involvement and importance of NRF2 in the context of viral infection.

## 1. Introduction

Nuclear factor erythroid 2-related factor 2, also known as NRF2, is a transcription factor initially discovered in 1994 as a key regulator of redox homeostasis in animal cells [1]. During homeostasis, NRF2 is constitutively expressed but kept inactive in the cytoplasm by Kelch-like ECH-associated protein 1 (KEAP1) [2]. Here, KEAP1 forms a complex with the E3 ubiquitin ligase CULLIN3 (CUL3) which promotes polyubiquitination of NRF2. This polyubiquitination subsequently leads to proteasomal degradation of NRF2, maintaining NRF2 levels and thus activity, at low levels at homeostasis [3]. Upon exposure of KEAP1 to electrophiles or reactive oxygen species (ROS), proteasomal degradation of NRF2 is abrogated and intracellular NRF2 levels increase. Importantly, increased NRF2 levels in the nuclear compartment leads to increased induction and regulation of NRF2 target genes which includes genes involved in the antioxidant response and in phase I, II, and III detoxification processes [4]

Although KEAP1 is now recognized as a central regulator of NRF2 levels, there are several other processes that contribute to either NRF2 activation or inactivation and some are independent of KEAP1. One such process is the heterodimerization of NRF2 with either of the small musculo-aponeurotic fibrosarcoma (sMAF) protein factors MafG, MafK, or MafF [4,5,6,7]. Although it is still not clear when and how one sMAF factor is preferred over the others, it is likely that each specific sMAF factor yields engagement with specific NRF2 target genes depending on the condition of the cell [4,8].

One common feature of the NRF2 target genes is an antioxidant response element (ARE) placed in proximity to the promotor region of NRF2 target genes [4]. Binding of the NRF2 heterodimer to these elements are, however, in some cases blocked by the binding of BTB and CNC homology 1 (Bach-1) to the ARE-element in a KEAP1-independent manner. Bach1 is a heme sensor and dissociates from the DNA in response to high intracellular heme levels. Bach1 dissociation then allows NRF2 to bind and induce genes such as heme oxygenase 1 (HO-1) to assist in restoring heme homeostasis [9]. A series of other inducing and regulatory mechanisms have also been described, although not to the same level of detail as the sMAFs and BACH-1. One such is the phosphatidylinositol-3-kinase and protein kinase B (PI3K-AKT) signaling pathway. Here, phosphorylation of glycogen synthase kinase 3 (GSK3) by AKT leads to additional phosphorylation of cytosolic NRF2 to further promote its proteasomal degradation through a CULLIN1 (CUL1)-dependent mechanism [10]. Furthermore, NRF2 can be activated through a p62-dependent mechanism. In this case, p62 promotes the release of NRF2 through direct interaction with KEAP1 [11]. NRF2 can also become activated through phosphorylation of the redox-sensitive protein kinase C (PKC), which also promotes NRF2 dissociation from KEAP1 and increased transcription of ARE-driven antioxidant genes [12].

Lately, a significant number of studies have demonstrated that NRF2 is central to processes outside the antioxidant response. For example, NRF2 has now been identified as a key regulator of metabolic shifts in cancer cells. Here, NRF2 promotes cancer development by redirecting glucose and glutamine into anabolic pathways to allow for cell proliferation [13]. This is supported by reports on NRF2 being involved in the regulation of metabolism in general and energy fluxes such as lipogenesis and amino acid transportation in the cell. For these reasons, NRF2 is now recognized as a master regulator of metabolic reprogramming under stress conditions (recently reviewed in [14]).

Furthermore, in recent years NRF2 has been demonstrated to be important both in responses to and for the protection from infection. Stimulation of immune cells with Toll-like receptor (TLR) ligands such as lipopolysaccharide (LPS) results in NRF2 activation, which reduces the release of proinflammatory cytokines [15]. Further, a link between NRF2 and antiviral responses has been made through its inhibition of central signaling components of interferon-inducing pathways including Stimulator of Interferon genes (STING) and Mitochondrial antiviral signaling (MAVS) [15]. In addition, and quite surprisingly, considering the regulatory effect on interferon (IFN), activation of NRF2 seems to have broad antiviral potential, including suppression of viral replication in Severe Acute Respiratory Syndrome-Corona Virus (SARS-CoV2), Zika virus, and Herpes Simplex virus infections in vitro [16]. In other cases, however, NRF2 activation seems pro-viral by promoting cell survival, suggesting that some viruses have evolved to exploit NRF2 to promote replication [17]. This review aims to summarize NRF2′s role during viral infection.

## 2. Reported Cases of NRF2 Activation during Viral Infections

Several studies document activation of NRF2 during infection with a wide variety of viruses through either direct or indirect mechanisms. One of the best described examples of direct viral induction of NRF2 comes from a study by Audrey Page et al. [18], who studied the mechanism of NRF2 activation during infection with Marburgvirus (MARV) in the human kidney cell line 293T. The authors demonstrated that VP24, a viral protein of Marburgvirus, is capable of binding to KEAP1 to induce the release of NRF2. The authors further demonstrated the activation of NRF2 using a reporter gene assay in combination with the overexpression of exogenous VP24, suggesting that VP24 expression alone is sufficient to drive NRF2 activation. Additionally, using Western blot analysis, the authors showed that NAD(P)H Quinone Oxidoreductase 1 (NQO1), a NRF2 downstream effector, was expressed in MARV infected Vero E6 cells.

Together, these data suggest that MARV VP24 can activate NRF2 through direct interaction with its inhibitor KEAP1 in vitro. The effects of NRF2 activation are not yet fully understood, but through in vivo experiments the group demonstrated a higher survival rate in Nrf2 knockout mice when compared to wild type mice [18]. Megan R Edwards et al. [19] made a similar finding in MARV-infected cells of the human monocytic cell line (THP1). Here, co-immunoprecipitation assays confirmed the ability of VP24 to directly interact with KEAP1. Additionally, the authors showed that VP24 expression induced the expression of the Nrf2 target genes NQO1, glutamate-cysteine ligase modifier subunit (GCLM) and HO-1. Accordingly, these reports provide evidence of direct interaction between VP24 and KEAP1, suggesting that MARV has developed a mechanism to engage the cellular antioxidant response as part of the replication strategy [19].

Evidence of indirect NRF2 activation during viral infection is more frequent but it has been difficult to determine if NRF2 activation is intended by the virus or just part of a protective host response to infection. As an example hereof, a study by Roberta Mastrantonio et al. [20] aimed to increase the understanding of why neurological complications are observed in approximately 50% of patients in the late phase of Human Immunodeficiency Virus (HIV) infection. The authors found that the HIV transcription factor Tat activates NRF2 in neuroblastoma cell lines through the increase of ROS. Here, the neuroblastoma cells were not directly infected by the virus: instead, NRF2 activation was triggered by increased levels of ROS released by HIV infected macrophages or glia cells in a Tat-dependent manner. Thus, in this case, NRF2 was activated as part of a bystander response to infection. It is still unclear if the induction of NRF2 affects viral propagation or neuronal pathology in these models of HIV infection [20]. In a study on Respiratory Syncytial Virus (RSV), Tao Sun et al. [21] reported the induction of NRF2 during in vitro infection of a human alveolar basal epithelial cell line (A549). Here, the authors demonstrated increased expression levels of NRF2 and of its downstream effector HO-1 24-hour post infection (HPI). In this case, the mechanism underlying NRF2 activation remains unknown, but the authors demonstrated that activation of NRF2 led to upregulation of TLR7 expression. As TLR7 is known to be part of the antiviral response through the induction of type I IFNs, this could be part of a host defense mechanism [21]. An additional study by Komaravelli et al. [22] also demonstrated activation of NRF2 in response to infection with RSV. In both studies, NRF2 activation was induced early in infection, but while Tao Sun et al. observed sustained NRF2 activation late in infection, Komaravelli et al. observed a decrease in NRF2 induction at 15HPI and 24HPI. Tao Sun et al. gave some explanation that could help to understand this major difference in the result with the Komaravelli study, which included differences in cell culture medium and in the multiplicity of infection (MOI) used in the two studies. However, both studies suggest that activation of NRF2 has a protective role in the response to virus-induced oxidative stress [22]. In a different study, activation of NRF2 was observed during infection with Severe Fever with Thrombocytopenia Syndrome Virus (SFTSV) in vivo and in vitro with human kidney cell lines (HEK293). Here, Younho Choi et al. [23] used mass spectrometry to identify a nonstructural protein of SFTSV that interacts with tripartite motif-containing protein 21 (TRIM21). This interaction facilitated p62-mediated KEAP1 sequestration and subsequently the release of NRF2 from KEAP1. This triggered the activation of NRF2, leading to the transcription of antioxidant genes including HO1, Nicotinamide Adenine Dinucleotide Phosphate Hydrogen (NAD(P)H), NQO1, and Cluster Determinant 36 (CD36). Furthermore, mutational analysis demonstrated that the nonstructural viral protein is necessary to increase NRF2 activity during infection. Interestingly, the mutated SFTSV nonstructural protein showed less cytopathogenic effect. Whether the activation of NRF2 is intended by SFTSV or whether it is an unintended bystander response to the interaction with TRIM21 and p62 is yet to be determined. Further investigations could focus on determining if the effect of NRF2 activation on viral propagation could be experimentally separated from the effect of TRIM21 and p62 on replication [23]. A similar indirect activation of NRF2 was found by Alexander V. Ivanov et al. [24]. Here, it was demonstrated that Hepatitis C Virus (HCV) activates the NRF2/ARE pathway in human liver cell line (HUH7) using luciferase reporter assay and expression on two NRF2 downstream effectors, HO1 and NQO1. They further highlight that the NRF2/ARE pathway is activated through several independent mechanisms mediated by the HCV core proteins. One such mechanism of NRF2 activation is triggered by PKC in response to accumulation of ROS. The authors also found that the effect of several viral proteins was mediated through the Casein Kinase 2 (CK2) and the PI3K. Finally, the study also showed that NRF2 can be activated independently of ROS production during HCV infection [24]. Involvement of PKC in NRF2 activation was also suggested by Stephanie Schaedler et al. [25] who demonstrated using a reporter gene assay that Human Hepatitis B Virus (HBV) induces a strong activation of NRF2/ARE-regulated genes in vitro in HepG2 cells. To go further, the authors investigated which viral proteins could activate NRF2. For this purpose, they used an expression vector of the viral proteins HBx and LHBs. Here, they observed an induction of the ARE reporter. These data suggest that the activation of NRF2 is induced by HBV proteins [25]. Interestingly, another study showed the implication of p62-KEAP1 interaction in NRF2 activation during the HBV infection. Here, Bo Liu et al. [26] confirmed the previous results that HBV stimulates NRF2 activation in hepatocyte cell lines (HepG2, HepG2.2.15, and HUH7 cell) and in human liver tissue from patients with chronic HBV infection. Additionally, they demonstrated the involvement of the p62-KEAP1 interaction in this mechanism, as the knock down of p62 reduced NRF2 activation by HBV [26]. In these cases, the link between p62 and c-Raf/MEK was not established. In 2013, Junsub Lee et al. [27] published a study showing NRF2 as increased in primary human foreskin fibroblasts (HFFs) infected with Human Cytomegalovirus (HCMV, Town strain). The authors showed an increase in NRF2 protein levels in the nucleus 24-hour post infection, which correlated with the induction of HO-1 expression. Furthermore, the authors investigated the role of CK2 as previously reported [28,29]. Using a CK2 inhibitor called 4,5,6,7-tetrabromobenzotriazole (TBB), the group observed that inhibition of CK2 activity prevented the increase in HO-1 expression and thus concluded that CK2 activity was required for HCMV-mediated NRF2 activation [27].

One thoroughly described viral mechanism of NRF2 activation is that of Kaposi’s Sarcoma-associated Herpes Virus (KSHV) described by Olsi Gjyshi et al. in three individual papers [30,31,32]. In one paper [30], the authors showed that during the de novo infection of HMVEC-d cells, NRF2 is activated through a mechanism dependent on ROS accumulation. Interestingly, the inhibition of NRF2 at an early stage of infection decreases the viral induction of VEGF-A, VEGF-D, and COX-2, which are known to be important in KSHV pathogenesis. The authors also demonstrated that at this early stage of infection, activated NRF2 is interacting with the viral DNA [30]. In the following two papers [31,32], the group investigated the latent phase of KSHV infection in long-term-infected telomerase-immortalized endothelial cells (TIVE-LTC). These studies showed that two simultaneous mechanisms of NRF2 activation are occurring during the latent phase of KSHV infection. The first mechanism of NRF2 activation was observed to be ROS-independent and occurring through the inhibition of autophagy, resulting in the accumulation of p62 and thus, a complete release of NRF2 from its interaction with KEAP1. The second mechanism stemmed from an association between NRF2 and the KSHV genome. Here, NRF2 colocalizes with the KSHV genome in a manner that is dependent on the latency-associated KSHV protein LANA-1, which allows for the transcription of the latent viral proteins. These data demonstrate that KSHV exploits NRF2 during the different phases of the viral cycle, through several independent mechanisms. This promotes the expression of the viral promoter Open Reading Frame (ORF) 50 during the early lytic burst and the expression of the viral promoter ORF73 during the latent infection phase. Moreover, the authors discovered two additional mechanisms depending on COX-2 to maintain high NRF2 activity during KSHV infection. These three studies of Olsi Gjyshi et al. demonstrate how complex NRF2 regulation supports KSHV replication [30,31,32]. Activation of NRF2 was also demonstrated for another member of the Herpes virus family by Emanuel Wyler et al. [33] and his colleagues investigated how Herpes Simplex Virus 1 (HSV1) infection affects the host cell transcriptome in early-stage infection, and vice versa. The authors showed that 5HPI, NRF2 is activated in primary human dermal fibroblasts. No underlying mechanism was described, therefore further investigation would be needed here [33]. In the study of Beata Kosmider et al. [34], the authors infected alveolar type II cells (ATII) and alveolar macrophages (AM) isolated from the lungs of patients with Influenza A Virus (IAV) infection. The authors showed that IAV infection increases ROS production, activates NRF2, and upregulates the NRF2 downstream effector HO-1 at both mRNA and protein level 48h after IAV infection. These data suggest that Influenza A virus infection induces activation of NRF2, most likely through the induction of ROS [34]. In a study dedicated to investigating the importance of cellular redox balance on pathogenicity of Tick-Borne Encephalitis virus (TEB-virus), Yulia V. Kuzmenko [35] and colleagues transfected HEK293T cells with a plasmid encoding the TBE nonstructural protein (NS1) and measured the production of ROS. These experiments yielded a high increase in ROS formation. Moreover, an increase in NRF2 activity was highlighted using an ARE reporter plasmid and by measuring expression of NQO1 and HO-1. These data suggest activation of the Nrf2/ARE pathway by TBE virus NS1 protein in HEK293T cells, again through the increase in ROS formation [35]. Finally, David Olagnier and his colleagues [36] showed that dengue virus (DENV) infection in primary differentiated dendritic cells activates NRF2, which subsequently induces the expression of its downstream effectors Heme Oxygenase-1 (HMOX-1), superoxide dismutase 2 (SOD2), NQO1, Glutamate-Cysteine Ligase catalytic subunit (GCLC), and GCLM [36]. Yi-Lin Chen et al. [37] confirmed this observation in murine monocytic cells (RAW264,7) while also determining the underlying cellular mechanism. Through Western blotting analysis, he showed that the protein levels of Protein kinase R-like ER kinase (PERK) were increased during Dengue infection. PERK is a protein kinase resident in the endoplasmic reticulum (ER) and is associated with the response to ER stress. The group showed that the use of ER stress chemical inhibitors (4-PBA) and PERK chemical inhibitor (GSK2606414) reduced Dengue virus-induced ARE activation. Together, these data suggest that Dengue virus activates NRF2 in a manner that depends on the activation of PERK [37]. In conclusion, NRF2 can be activated thought multiple mechanisms during viral infection. In some cases, NRF2 activation is induced through direct viral interaction while in other instances NRF2 activation seems to be the result of cellular responses to disturbances in cellular homeostasis induced by viral replication. Please find in the Table 1 a list of the viruses presented in this section and in Figure 1 a schematic representation.

## 3. Reported Cases of NRF2 Downregulation during Viral Infections

In support of the notion that NRF2 is in many cases antiviral, several viruses seem to target either the expression or the functionality of NRF2. A recent example of this was highlighted by us and collaborators [16], where we found NRF2-inducible genes to be suppressed in lung biopsy samples obtained from patients with severe SARS-CoV-2 infection. Although the mechanisms employed by SARS-CoV2 to target NRF2 are not identified, it seems likely that inhibition of NRF2 could be an important viral tactic to ensure efficient propagation [16]. More mechanistic insight is provided in the case of inhibition of NRF2 by Enterovirus 71 (EV71). Here, Zhenzi Bai et al. [38] showed that infection with EV71 stimulates accumulation of ROS while still suppressing the activation of NRF2 and thus induction of its downstream effector HO-1. By contrast, the expression of KEAP1 was increased at both mRNA and protein level, suggesting that EV71 targets NRF2 through induction of its inhibitor [38]. Another recent example of the mechanisms of NRF2 downregulation is given by Patra et al. [39] who showed in African green monkey fetal kidney cell line infected with simian rotavirus RV-SA11 that rotavirus infection also downregulated NRF2. Interestingly, NRF2 downregulation occurs after an initial upregulation of NRF2 triggered by the oxidative stress generated by the viral replication. Moreover, after 3 h post infection, the authors observed by western blotting that NRF2 protein level and target genes such as HO-1, NQO1, and SOD1 were downregulated. Moreover, using the proteasome inhibitor MG132, Patra et al. demonstrated the importance of the proteasome in NRF2 downregulation during RV infection, suggesting that RV infection downregulates NRF2 through the proteasomal degradation [39]. Another virus suppressing NRF2 activity is HCV, as demonstrated in human hepatocytic cell lines by Monica Carvajal-Yepes and her colleagues [40]. Interestingly, the authors showed that sMaf proteins delocalized from the nucleus to the perinuclear region during infection, thus possibly impairing the sMaf-mediated expression of ARE gene by NRF2. Moreover, the authors used different HCV replicon constructs to demonstrate that the delocalization of sMaf were regulated by the HCV core proteins. These data suggest that the inhibition of ARE gene expression is driven by HCV core protein through the delocalization of sMaf, leading to the impaired function of NRF2. This phenomenon was also observed in primary liver tissue samples collected from HCV infected patients [40]. According to Alexander V Ivanov et al. [24], this could be an example of differential NRF2 regulation during the early and late stage of HCV infection. Indeed, as presented previously, Ivanov et al. showed that HCV core proteins activate NRF2 in the early stages of the infection. The authors suggest that in the early stages, the infected cells might benefit from NRF2 activation by protecting themselves against the oxidative stress generated by viral infection. However, at the late stage of infection, NRF2 activity might be unfavorable for the virus, and HCV therefore uses the strategy to reduce NRF2 activity [24], as also showed by Monica Carvajal Yepes et al. [40].

There are, however, also conflicting results in this area. Indeed, in opposition to the study by Roberta Mastrantonio et al. [20], which reported that HIV Tat protein induced activation of NRF2 in neuronal cells, a study by Bashar S Staitieh et al. [41] demonstrated that the viral proteins Tat and gp120 suppressed NRF2 in human monocyte derived macrophages. Both HIV proteins seemed to cause reduced levels of NRF2 and reduced the expression of its downstream effectors NQO1 and GCLC. Thus, the authors concluded that HIV factors Tat and gp120 repress NRF2 in macrophages, and that this led to innate immune dysfunction. These seemingly contradictory studies point to a possible difference in how HIV affects NRF2 in different tissues and cell types [41]. As briefly mentioned in the previous section, RSV was described as a suppressor of NRF2 in later stages of infection in both small alveolar epithelial cells (SAECs) and in human alveolar basal epithelial cell line (A549) [22]. The authors further demonstrated that lactacystin, a selective inhibitor of the proteasome, restored NRF2 cellular levels and expression of NRF2-inducible genes like NQO1 and Antioxidant Enzyme (AOE). Moreover, Narayana Komaravelli et al. went on to discover that RSV infection is associated with NRF2 deacetylation, an effect that could be completely restored by the use of Trichostatin A, a chemical inhibitor of a histone deacetylase. Finally, Narayana Komaravelli et al. observed a decrease in basal NRF2 acetylation and a reduction in NRF2 levels in the nucleus in lung tissue of mice infected with RSV. These findings highlight a mode of action in which RSV induces NRF2 activity in early stages of infection before significant suppression of NRF2 leading to oxidative stress both in vitro and in vivo [22]. In a follow-up study, Narayana Komaravelli et al. described a potential mechanism for the RSV-induced deacetylation, ubiquitination, and proteasomal degradation of NRF2 [42]. In this second study, the authors showed that the observed degradation of NRF2 occurs in a SUMO-specific E3 ubiquitin ligase–RING finger protein 4 (RNF4)-dependent manner. They move on to demonstrate this to be KEAP1 independent as siRNA mediated knock down of KEAP1 still allow RSV infection to downregulate NRF2. To further demonstrate this concept, the authors performed siRNA mediated knock down of RNF4 and showed by Western blotting analysis that nuclear NRF2 levels were restored when RNF4 was silenced in RSV infected cells. Moreover, the authors showed the importance of the membrane free subnuclear compartments named promyelocytic leukemia protein-nuclear bodies (PML-NBs) in the degradation process of NRF2. The PML protein is a member of the TRIpartite Motif (TRIM) family and a major component of PML-NBs complex. PML-NBs are involved in posttranslational modification such as SUMOylation. The authors used confocal microscopy analysis to demonstrate that RSV infected cells have increased levels of PML. To go further, the authors silenced PML expression with siRNA and showed decreased NRF2 degradation in PML silenced cells. All together these data indicate that PML-NBs are involved in RSV-mediated NRF2 degradation and that inhibition of PML-NBs formation rescues NRF2 nuclear levels and NRF2 downstream effectors. Another case of Nrf2 downregulation during virus infection is brought by Ai et al. [43] and their study on Coxsackievirus B3 (CVB3), an enterovirus associated with viral myocarditis. Here, the authors showed that during CVB3 infection, Nrf2 mRNA and protein levels were down regulated in vitro with infected mice cardiomyocytes and in vivo with mice heart tissues. Moreover, the authors showed that HO-1 protein level and the concentration of the antioxidant enzymes glutathione peroxidase (GSH-Px) and superoxide dismutase (SOD) were also reduced in the cardiac tissue of CVB3-infected mice. Finally, the authors performed pulmonary RSV infection in mice, observing SUMOylation of NRF2 in vivo. This significatively reduced the nuclear levels of NRF2, suggesting a similar mechanism exists both in vitro and in vivo [42]. Finally, another in vivo example of Nrf2 suppression during virus infection comes from Rabbit Hemorrhagic Disease Viruses (RHDV) reported by Bo hu et al. [44]. In this study, the authors used proteomic analysis on rabbit liver tissues to identify differentially regulated proteins after RHDV infection. Analysis of liver tissues from the infected rabbits showed that infection with RHDV led to decreased expression of ARE-regulated genes. Moreover, immunofluorescence and western blotting experiments showed an increased translocation into the nucleus of the complex KEAP1-Nuclear factor-kappa B (NFκB) triggering the nuclear export of Nrf2 and subsequent degradation during RHDV infection. Together, these findings highlight Nrf2 suppression during RHDV infection as being triggered by the nuclear export of Nrf2 by the KEAP1-NFκB complex [44]. Please find in the Table 1 a list of the viruses presented in this section and in Figure 1 a schematic representation.

## 4. Importance of NRF2 in Viral Infections

Although examples exist where specific species of virus highjack NRF2 as an integrated part of replication, recent lines of evidence promote the notion that in general, NRF2 activation is protective during viral infection. The protection can be mediated either through antiviral effects by inhibition of viral replication, either through inhibition of cell death to protect from excessive tissue damage, or both. In this session, we will outline current knowledge of the protective effect of NRF2.

### 4.1. Protective Role of NRF2

One of the first studies demonstrating a protective effect of NRF2 activation during virus infection was published in 2008 by Hye Youn Cho et al. [45]. Here, using a murine model of Respiratory Syncytial virus (RSV) infection, the authors could observe delayed viral clearance in the bronchoalveolar lavage (BAL) and an increase in bronchoalveolar injury in Nrf2^−/−^ mice in comparison with control mice. Furthermore, authors observed an increase in neutrophilic and eosinophilic infiltration into the lung tissue of the Nrf2 deficient mice. The authors quantified the oxidation of endogenous macromolecules and observed an increase in protein oxidation by 20–25% in infected Nrf2^−/−^ mice in comparison with control mice Nrf2^+/+^. According to the authors, these data suggest that in addition to its direct antiviral effect, Nrf2 plays a protective role against oxidation and inflammation generated by RSV infection [45]. Another example of the cytoprotective effect of NRF2 during viral infection was provided by Beata Kosmider et al. [34], who showed that the activation of NRF2 decreases the percentage of necrotic human alveolar epithelial cells during Influenza A viral infection. The observed effect of NRF2 on viral replication was modest (less than twofold reduction), whereas the NRF2-mediated reduction in necrosis seemed to be quite strong, indicating that the effect of NRF2 on necrosis was mediated mostly through the direct cytoprotective function of NRF2 [34]. Two studies have, however, reported significant antiviral effects of NRF2 activation on IAV replication. Matthew J Kesic et al. [46] showed that overexpression of NRF2 in primary human nasal epithelial cell reduce the viral entrance of IAV. Adding to this, Masaki Shoji et al. [47] used the Nrf2 activator Bakuchiol in Madin Darby Canin Kidney (MDCK) cells infected with IAV. The authors showed that Nrf2 activation with Bakuchiol inhibited the IAV mRNA and protein level and increased the survival rate of infected cells [47]. Together, these data showed an antiviral effect of NRF2 activation on IAV infected cells. Since these initial discoveries have been made, the list of viruses sensitive to NRF2 activation has kept increasing. To investigate the importance of NRF2 driven genes during infections with Ebola virus, Hanxia Huang et al. [48] used a chemical compound called Hemin as an activator of the NRF2-driven gene HO-1. In this study, monocytes isolated from human peripheral blood mononuclear cell, HeLa cells and HFF1 cells were treated with Hemin and then infected with Ebola virus (EBOV). The authors found that treatment with Hemin efficiently reduced viral replication of Ebola virus. These data lead the authors to conclude that the activation of HO-1 has an antiviral effect on EBOV [48]. Hanxia Huang et al. also investigated the antiviral effect of Hemin in another study [49] with primary human macrophages infected by Zika virus (ZIKV). The authors showed that Hemin treatment decreases Zika virus mRNA. They further observed this decrease was happening in a NRF2 dependent manner as silencing of NRF2 diminished the antiviral effect of Hemin against Zika infection. These data together show that Hemin also has an antiviral effect against ZIKV [49]. Lately, multiple studies investigating the use of NRF2 activators as a broad-spectrum approach to antiviral treatment has been performed in vitro with human cells. One such study was performed by Upayan Patra et al. [50] who demonstrated an antiviral effect of three pharmacological activators of Nrf2/ARE pathway—RA-839, 2-cyano-3,12-dioxo-oleana-1,9(11)-dien-28-oic acid methyl ester, and Hemin—during in vitro rotavirus (RV) infection [50]. The authors showed that Nrf2 activation led to decreases of both RV RNA and protein expression thus having strong effects on RV replication. Interestingly, the authors showed that this antiviral effect is independent of interferon stimulation [50]. Other studies quickly added more evidence to the antiviral potential of NRF2 activation. As an example, Zhenzi Bai et al. [38] showed that silencing NRF2 is beneficial for viral replication of enterovirus EV71 and that activating NRF2 through genetic silencing of KEAP1 reduced viral replication in Rhabdomyosarcoma (RD) cells. These results suggested that NRF2 downregulation is required for efficient EV71 propagation and suggest that activation of NRF2 could be antiviral [38]. Two additional studies used an in vivo approach confirming in vitro observations that chemical activation of NRF2 is antiviral. Ruth Seelige et al. used a mouse model comparing Nrf2^−/−^ mice with wild type mice during infection with murine cytomegalovirus (MCMV) [51]. The authors studied the recruitment of immune cells into infected tissues and found that early immune cell accumulation MCMV infected mice was dependent on Nrf2. Mice deficient in Nrf2 showed a decreased number of innate immune cells in the peritoneal region and were more susceptible to MCMV infection than the control mice. Moreover, the authors could partially induce innate leukocyte infiltration in the peritoneal tissue by applying the Nrf2 activator tert-butylhydroquinone (tBHQ) Interestingly, this treatment increased protection against MCMV infection in the mice [51]. Bo Hu et al. [44] reported a case where up regulation of Nrf2 during RHDV infection in rabbits was beneficial for the host, as it reduces the death rate. Notably, induction of Nrf2 with the Nrf2 chemical activator tBHQ, prolongated rabbit life span after RHDV infection. These data hint to a protective role of Nrf2 in this model [44]. Recently, a study published by us together with several collaborators [16] demonstrates that treatment with the chemical NRF2 activators 4-octyl itaconate (4-OI) [52] or dimethyl fumarate (DMF) had strong anti-SARS-CoV-2 effects, suggesting a NRF2 driven antiviral program operating independently of type I Interferon. Importantly, the antiviral effect of NRF2 activation was preserved across multiple viruses including Vaccinia virus (VACV), HSV, and ZIKV. These conserved antiviral effects suggest that NRF2 controls a broadly acting antiviral program and thus could serve as a therapeutic target for virus during infection [16,53,54].

### 4.2. Pathogenic Role of NRF2

Although NRF2 has mostly been reported to be protective in viral infections, a few studies demonstrate a pathogenic role of NRF2. In 2014, three studies were published describing a pathogenic role of NRF2. One of these studies were performed by Audrey Page et al. [18], who studied the pathogenic impact of Nrf2 activation in vivo during MARV infection. Here, the authors showed that Nrf2-deficient mice displayed improved control of MARV infection as compared to wild type (WT) mice. In fact, while WT mice succumbed to infection by day 8, mice lacking Nrf2 demonstrated a 50% increase in survival rate. Moreover, Nrf2 deficient mice showed a lower viral titer and complete clearance of virus by day 9. This suggests a role by which Nrf2 contributes to MARV pathogenesis, although the full extent of the contribution has not yet been understood [18]. The second study was performed by Olsi Gjyshi et al. and showed the impact of NRF2 inhibition during in vitro infection with KSHV. The group found that inhibiting NRF2 during infection decreased induction of anti-KSHV host factors such as VEGF-A, VEGF-D and COX-2. Hereby, NRF2 activation seems to be beneficial for the virus as it contributes to the KSHV expansion [30]. The third study was published by David Olagnier et al. [36] and demonstrates silencing of NRF2 as an inducer of a strong antiviral response. This observation was coupled with the observation that NRF2 silencing allowed ROS-induced apoptosis, thus abolishing the protection stemming from the antiviral effects NRF2 [36].

An additional study highlighting NRF2-driven pathogenicity was performed by Tetsuya Saito et al. [55] in their study of HCV-positive hepatocellular carcinoma cells. Here, the authors demonstrate NRF2 activation during HCV infection which in turn leads to a metabolic shift. During this shift, glucose is redirected into the glucuronate pathway while glutamine is used for glutathione synthesis. In this model, the metabolic shift increases the proliferation of malignant cells; prompting the authors to conclude that the inhibition of NRF2 activation could reduce the proliferation of HCV infected cells and improve anticancer treatment. In this setting, NRF2 is not pro-viral as such, but promotes virus-induced cancer by changing cell metabolism [55]. Recently, in a study published by Younho Choi et al. [23], the group shows that NRF2 also contributes to the pathogenicity of the Severe fever seen during infection with SFTSV in mice. Indeed, the authors showed that a viral nonstructural protein of SFTSV triggered the activation of NRF2. Additionally, mice infected with a strain of SFTSV with a mutated nonstructural protein showed a higher survival rate and decreased weight loss. Interestingly, in this case the viral replication rate did not seem to be affected by the mutation in the nonstructural protein, suggesting that the increase of the pathogenicity observed when NRF2 is activated is not related to an increase in the viral replication [23]. These studies all highlight a dual role of NRF2 during viral infections, as NRF2 activation could contribute to viral expansion in some cases. Please find in the Table 1 a list of the viruses presented in this section and in Figure 1 a schematic representation. 

**Table 1 antioxidants-10-01491-t001:** Viruses and NRF2 among publications.

Virus	Model	Effect on NRF2	Viral Replication	Patho/Cytoprotective effect	Mechanism	Ref
HIV	In vitro expression of HIV protein tat in neuroblastoma cell line	↑	ND		Induction of ROS	[20]
	In vitro infection of primary human monocytes and in vivo rat	↓	ND		ND	[41]
RSV	In vitro with human alveolar basal epithelial cell line	↑	↓		ND	[21]
	In vivo with Nrf2^+/+^ and Nrf2^−/−^ mice	↑	↓		ND	[45]
	In vivo with mice and in vitro with human alveolar basal epithelial cell line	↓	ND		Proteasomal degradation of Nrf2	[22]
	In vitro with human alveolar epithelial cell line	↓	ND		SUMOylation and ubiquitination of Nrf2 induce its degradation	[42]
SFTSV	In vivo with mice	↑	ND	Pathogenic: mice infected with SFTSV mutated for this nonstructural protein showed a higher survival rate and less weight loss.	SFTSV viral protein interact with TRIM21 that normally mediated p62 ubiquitination. Hence p62 is stabilized and interact with KEAP1 leading to set free Nrf2	[23]
HCV	In vitro with hepatocyte cell line	↑	ND		HCV core proteins activate Nrf2/ARE pathway via several independent mechanisms. 1. Nrf2 activation was triggered by protein kinase C in response to accumulation of ROS. 2. Nrf2 was also activated in ROS independent manner 3. The effect of some viral proteins was mediated through casein kinase 2 and phosphoinositide 3 kinase.	[24]
	In vitro with hepatocyte cell line	↓	ND		HCV Core and nonstructural proteins induce nuclear export of sMaf a Nrf2 partner	[40]
	In vitro with hepatocellular carcinoma cell line	ND	ND	Pathogenic: Persistent activation of NRF2 makes HCV positive tumor cells resistant to oxidative damage and anticancer agent		[55]
HCMV	In vitro with primary human foreskin fibroblasts	↑	↓		Casein Kinase 2 mediated	[27]
HBV	In vitro with human hepatoma derived cell lines	↑	ND		Triggered by HBV proteins (HBx and LHBs)	[25]
	In vitro with human hepatocyte cell lines	↑	ND		HBV protein HBx stimulate p62-Keap1 interaction leading to Nrf2 activation	[26]
MARV	In vitro with human kidney cell line	↑	ND	Pathogenic: Infected mice deficient for Nrf2 have a higher survival rate than infected control mice	VP24 viral protein bind to KEAP1 setting free Nrf2	[18]
	In vitro with lymphocyte cell line	↑	↓	Cytoprotective: Activation of antioxidant mechanisms	MARV protein Mvp24 interact directly with KEAP1	[19]
KSHV	In vitro with Human dermal microvascular endothelial cell line	↑	ND	Pathogenic: Nrf2 contribute to the expression of host factor VEGF-A, VEGF-D and COX-2 important for KSHV pathogenesis	Induction of ROS	[30]
	In vitro with primary effusion lymphoma cell line and tissue	↑	↑		COX-2/PGE2 axis induces Nrf2 through prostaglandin E receptor 4 (EP4)	[32]
	In vitro with TIVE and LTC cells	↑	↑		Two distinct mechanisms: (1) Ros induction. (2) ROS-independent and through P62–Keap1 interaction	[31]
IAV	In vitro with alveolar type II cells and alveolar macrophages isolated from human lungs infected	↑	↓		ND	[34]
	Primary human nasal epithelial cell	ND	↓		ND	[46]
	In vitro Madin darby canine kidney cell line	ND	↓		ND	[47]
HSV1	In vitro with primary normal human dermal fibroblast cell line	↑	↓		ND	[33]
TBEV	In vitro with human embryonic kidney cell line	↑	ND		TBEV nonstructural protein induces ROS production	[35]
Broad	In vitro chemical activation of NRF2 before infection inhibited replication of HSV-1/2,VACV, ZKV, SARS-CoV2	↑	↓		ND	[16]
DENV	In vitro with murine monocytic cell line	↑	ND		DENV infection induces ER stress that phosphorylate PERK that activate Nrf2	[37]
	In vitro with monocyte isolated from human peripheral blood mononuclear cell from healthy donors	↑	ND		ND	[36]
RHDV	In vivo with rabbits	↓	ND	Cytoprotective: Upregulation of NRF2 reduce infected rabbit’s death rate	Nuclear export of Nrf2 by KEAP1-NFκB complex	[44]
EV71	In vitro with human rhabdomyosarcoma cell line	↓	↓		Up regulation of Keap1	[38]
CMV	In vivo with mice Nrf2^−/−^ and Nrf2^+/+^	ND	↓		ND	[51]
ZIKV	In vitro with primary human macrophages	ND	↓		ND	[49]
EBOV	In vitro with human cell lines	ND	↓		ND	[48]
RV	In vitro with human cell lines	ND	↓		ND	[50]
	In vitro with green monkey fetal kidney cell line	↓	ND		Proteasomal degradation dependency	[39]
CVB3	In vitro with mice cardiomyocytes and in vivo with mice	↓	ND	Contribute to oxidative stress		[43]

ND: not determined.

## 5. Conclusions

From initially being described primarily as a regulator of redox homeostasis and an important contributor to chemoresistance in many types of cancer, NRF2 is now placed as one of the central players in regulating immunity, including immunity to infection. NRF2 involvement has been demonstrated for a wide range of different viruses, and although some viruses have evolved to exploit NRF2 for replication, it is in most cases inducing antiviral immunity. Despite the clear demonstrations of the antiviral properties of NRF2, it is not clear which NRF2 controlled genes are mediating these effects. A few publications have identified some of the anti-viral mediators (e.g., HO-1) but it is still not very clear how the effectors actually inhibit the infection. NRF2 generally has cytoprotective functions, which can protect against virus-induced cell death and subsequent inflammation-induced immunopathology. However, in many cases, the antiviral properties and the anti-inflammatory/cytoprotective properties seem to be separate mechanisms. We suspect that focus in the coming years will be on characterizing the molecular mechanisms underlying the NRF2-induced antiviral program.

## Figures and Tables

**Figure 1 antioxidants-10-01491-f001:**
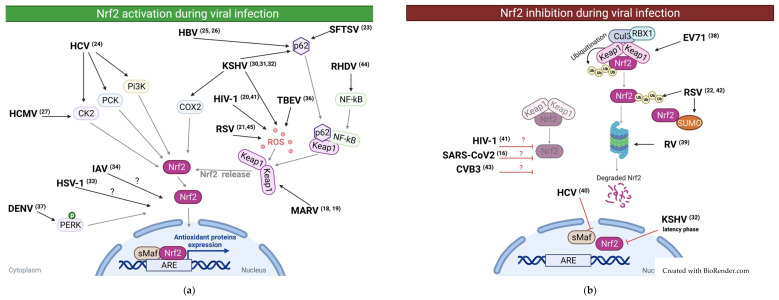
Graphical abstract of effect observed in the literature on NRF2 during viral infections. (**a**) Graphical overview of the activation of NRF2 during viral infection. (**b**) Graphical overview of the inhibition of NRF2 during viral infection.

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
