# Peer review of "NRF2 in Viral Infection"

_antioxidants, 2021, doi:10.3390/antiox10091491_

Round 1

Reviewer 1 Report

This is overall a well written review. As an updated to a recently published review on a similar topic, the authors should add a few more papers there were not included in the current manuscript. These are listed below. Reference 12 in Table 1 for Zika virus is incorrectly cited.

Coxsackievirus B3 is an enterovirus that is associated with viral myocarditis. Ai et al. (2017) showed that Nrf2 mRNA and protein levels, HO-1 protein levels, and the activities of the antioxidant enzymes GPx and SOD were reduced in coxsackievirus B3-infected mice and cardiac myocytes.

Bai et al. (2020) reported that enterovirus EV71 infection stimulates ROS production, upregulates Keap1, and downregulates Nrf2 protein levels. EV71 infection also resulted in increased Nrf2 ubiquitination, which contributed to Nrf2 reduction. The findings suggested that Nrf2 downregulation is required for efficient EV71 propagation. EV71 virus reduces Nrf2 activation to promote production of ROS in infected cells, inflammatory reactions, and cell death, with a crucial effect on viral replication.

Patra et al (2020) described that rotavirus infection also affected the cellular antioxidant defense by rendering degradation of the Nrf2 via the ubiquitin-proteasome pathway. The Nrf2 protein levels decline with the progression of rotavirus infection (after an initial increase), along with lowered expression of Nrf2 target genes HO-1, NQO1, and SOD1 (Patra et al. 2020).

RA-839, a recently discovered small molecule with potent and highly selective agonistic activity towards cellular redox stress-sensitive Nrf2/Antioxidant Response Element pathway, inhibits rotavirus RNA and protein expression, viroplasm formation, yield of virion progeny and virus-induced cytopathy.(Patra et al 2019)

Author Response

Dear Reviewer #1

We are pleased with the overall endorsement of our manuscript. Please find below a point-by-point reply to your suggestions for revision.

This is overall a well written review. As an updated to a recently published review on a similar topic, the authors should add a few more papers there were not included in the current manuscript. These are listed below.

  • Reference 12 in Table 1 for Zika virus is incorrectly cited. Thank you a lot, I corrected the reference (48) in Table 1.
  • Coxsackievirus B3 is an enterovirus that is associated with viral myocarditis. Ai et al. (2017) showed that Nrf2 mRNA and protein levels, HO-1 protein levels, and the activities of the antioxidant enzymes GPx and SOD were reduced in coxsackievirus B3-infected mice and cardiac myocytes. Agreed, I add it in the session ”3. Reported cases of NRF2 down regulation during viral infections” Line 306-312 and page number 6
  • Bai et al. (2020) reported that enterovirus EV71 infection stimulates ROS production, upregulates Keap1, and downregulates Nrf2 protein levels. EV71 infection also resulted in increased Nrf2. ubiquitination, which contributed to Nrf2 reduction. The findings suggested that Nrf2 downregulation is required for efficient EV71 propagation. EV71 virus reduces Nrf2 activation to promote production of ROS in infected cells, inflammatory reactions, and cell death, with a crucial effect on viral replication. We agree with the reviewer that this is an importatant study and therefore the study is included in section ”3. Reported cases of NRF2 down regulation during viral infections”
  • Patra et al (2020) described that rotavirus infection also affected the cellular antioxidant defense by rendering degradation of the Nrf2 via the ubiquitin-proteasome pathway. The Nrf2 protein levels decline with the progression of rotavirus infection (after an initial increase), along with lowered expression of Nrf2 target genes HO-1, NQO1, and SOD1 (Patra et al. 2020). Really nice study, thank you for suggesting it. I have added to section 3. Reported cases of NRF” down regulation during viral infections” ”Line 241-249 page 5
  • RA-839, a recently discovered small molecule with potent and highly selective agonistic activity towards cellular redox stress-sensitive Nrf2/Antioxidant Response Element pathway, inhibits rotavirus RNA and protein expression, viroplasm formation, yield of virion progeny and virus-induced cytopathy.(Patra et al 2019). Thank you for this suggestion. We agree that this study is important and is now included in section ”4.1 Protective role of NRF2”

Thank you for the references suggestions and constructive feedback.

Reviewer 2 Report

The authors reviewed the current knowledge about the role of NRF2 in viral infections. In the abstract and throughout the text, they mainly highlight the anti-viral effect of NRF2 while evidence suggests a dual role of NRF2 in virus infection (as they also mentioned). A better clarification why they think that NFR2 has mainly anti-viral effect when some viruses activate NRF2 during infection (as they summarized in 2. Reported cases of NRF2 activation during viral infections) should be presented.

Specific comments:

  1. In the Introduction, the authors should better explain NRF2 activation and inactivation. Distinguishing KEAP1-dependent and KEAP1-independent mechanisms could contribute to better clarity.
  2. In the Introduction, the authors state: “This review aims to summarize NRF2’s role during viral infection (recently reviewed here(17)).” Why is the aim of their review to summarize something that was recently reviewed elsewhere?
  3. The authors should provide a full name of NQO1 and not just Quinone Oxidoreductase 1 (NQO1).
  4. Heme Oxygenase-1 is mainly abbreviated as HO-1 but also as HMOX-1. Abbreviations should be uniform.
  5. Table 1 should be mention within the text also.
  6. References should be prepared according to the journal’s Author Instructions: “In the text, reference numbers should be placed in square brackets [ ],…”

Author Response

Dear Reviewer #2

We are pleased with the overall endorsement of our manuscript. Please find below a point-by-point reply to your suggestions for revision.

The authors reviewed the current knowledge about the role of NRF2 in viral infections. In the abstract and throughout the text, they mainly highlight the anti-viral effect of NRF2 while evidence suggests a dual role of NRF2 in virus infection (as they also mentioned). A better clarification why they think that NFR2 has mainly anti-viral effect when some viruses activate NRF2 during infection (as they summarized in 2. Reported cases of NRF2 activation during viral infections) should be presented.

We failed to identify where in section 2 the reveiwer is reffering to. However, in the start for section three we state that NRF2 is mainly anti-viral. We agree that this statement is perhaps too bold. For this reason, we have modified the text slightly.

Specific comments:

  1. In the Introduction, the authors should better explain NRF2 activation and inactivation. Distinguishing KEAP1-dependent and KEAP1-independent mechanisms could contribute to better clarity.

I have added line 30 and 39 that the mechanisms were KEAP1 independent.

  1. In the Introduction, the authors state: “This review aims to summarize NRF2’s role during viral infection (recently reviewed here(17)).” Why is the aim of their review to summarize something that was recently reviewed elsewhere?

That is a very good point. We have taken out this particular reference as the topic is not completely overlapping.

  1. The authors should provide a full name of NQO1 and not just Quinone Oxidoreductase 1 (NQO1).

Yes thank you, I changed it for NAD(P)H Quinone Oxidoreductase 1 (NQO1)

  1. Heme Oxygenase-1 is mainly abbreviated as HO-1 but also as HMOX-1. Abbreviations should be uniform.

Ok thank you, I made the distinction between the gene HMOX-1 and the protein HO-1.

  1. Table 1 should be mention within the text also

We agree with the reviewer that the table must be called out in the text. We have now included this in section 2, 3, and 4.

  1. References should be prepared according to the journal’s Author Instructions: “In the text, reference numbers should be placed in square brackets [ ],…”

Done, thank you.

Thank you a lot for your help in finding the typing mistakes and thank you for your critical feedback about our approach and aim.

Reviewer 3 Report

This review article described pathophysiological interaction between stress responsive transcription factor Nrf2 and viral infection. Nrf2 plays an important role in cells including maintenance of redox homeostasis, drug metabolism through its target gene induction. In addition, Nrf2 also exhibits various function such as anti-inflammatory function. Curiously, virus infection modulates Nrf2 activity positively or negatively according to the virus species as the authors summarized in the manuscript. The authors also documented the protective and pathological effects of Nrf2 pathway in virus-infected organisms. 
Frankly speaking, the manuscript is well written and properly organized. The reviewer thinks the manuscript would provide interesting informations for the readers of Antioxidant and be acceptable with minor revisions described below. 

1) Line 85, Line 132 & Page 18; The abbreviation of NQO1 should be revised. 
Line 131  “Nicotinamide Adenine Dinucleotide Phosphate Hydrogen (NAD(P)H), NQO1” would be “NAD(P)H  quinone oxidoreductase 1 (NQO1) ”. 
Page 18, NQO1 is duplicated.

2) Line 94, Line 222;  GCLM should be “Glutamate-Cysteine Ligase modifier subunit”, and GCLC is “Glutamate-Cysteine Ligase catalytic subunit”. Glutamate-Cysteine Ligase is composed of GCLC/ GCLM heterodimer.

3) Line 116, The sentence of “Two additional studies...” looks strange. Ref. 21 is documented in the previous sentence. .

4) Line 153, Ref 26 would be incorrect.

5) Line 169, Junsub Lee et al.(27) would be Ref 30.

6) Line 214, “in HEK293T cells” is duplicated in the sentence.

7) Line 265 Add Ref. number for Roberta Mastrantonio et al. 

8) Line 385, tBHQ is abbreviated avobe.

9) Page 18, Line 1, “antioxydant” would be a typo.

Author Response

Dear Reviewer #3

We are pleased with the overall endorsement of our manuscript. Please find below a point-by-point reply to your suggestions for revision.

This review article described pathophysiological interaction between stress responsive transcription factor Nrf2 and viral infection. Nrf2 plays an important role in cells including maintenance of redox homeostasis, drug metabolism through its target gene induction. In addition, Nrf2 also exhibits various function such as anti-inflammatory function. Curiously, virus infection modulates Nrf2 activity positively or negatively according to the virus species as the authors summarized in the manuscript. The authors also documented the protective and pathological effects of Nrf2 pathway in virus-infected organisms. 
Frankly speaking, the manuscript is well written and properly organized. The reviewer thinks the manuscript would provide interesting informations for the readers of Antioxidant and be acceptable with minor revisions described below. 

1) Line 85, Line 132 & Page 18; The abbreviation of NQO1 should be revised.
Line 131  “Nicotinamide Adenine Dinucleotide Phosphate Hydrogen (NAD(P)H), NQO1” would be “NAD(P)H  quinone oxidoreductase 1 (NQO1) ”. -> thank you, I changed it.
Page 18, NQO1 is duplicated. I could not find where NQO1 was duplicated page 18, however I found it duplicated in the abbreviation list, so I corrected it.

2) Line 94, Line 222;  GCLM should be “Glutamate-Cysteine Ligase modifier subunit”, and GCLC is “Glutamate-Cysteine Ligase catalytic subunit”. Thank you I changed it.

3) Line 116, The sentence of “Two additional studies...” looks strange. Ref. 21 is documented in the previous sentence. .I do agree, thank you I changed it

Indeed it is a mistake, I changed by ”An additional study by Komaravelli et al.[22], also demonstrated activation of NRF2 in response to infection with RSV.”

4) Line 153, Ref 26 would be incorrect. -> yes i changed it thank you

5) Line 169, Junsub Lee et al.(27) would be Ref 30. -> thank you I changed it, now it is 28

6) Line 214, “in HEK293T cells” is duplicated in the sentence. -> thank you, I changed it

7) Line 265 Add Ref. number for Roberta Mastrantonio et al. -> thank you, I changed it

8) Line 385, tBHQ is abbreviated avobe. -> thank you, I changed it

9) Page 18, Line 1, “antioxydant” would be a typo -> thank you, I changed it

Thank you for your methodic approach and for your help finding the last typo.

Reviewer 4 Report

In this interesting review, the authors reported cases of NRF2 activation or down regulation during viral infections in vitro and in vivo. The protective role of NRF2 in viral infections is largely described. However, a limited number of studies demonstrate a pathogenic role of NRF2. Table and figure fit well with the objective of this review describing the multifaceted aspects of NRF2.

Author Response

Dear Reviewer #4

We are pleased with the overall positive feedback of our manuscript.

In this interesting review, the authors reported cases of NRF2 activation or down regulation during viral infections in vitro and in vivo. The protective role of NRF2 in viral infections is largely described. However, a limited number of studies demonstrate a pathogenic role of NRF2. Table and figure fit well with the objective of this review describing the multifaceted aspects of NRF2.

Round 2

Reviewer 1 Report

The authors have addressed the reviewer's suggestions

Author Response

Dear Reviewer

We have improved the English writing to accommodate your request.

Best,

Christian

Reviewer 2 Report

The authors improved their manuscript. Yet, references are still not prepared under the journal’s Author Instructions:

https://www.mdpi.com/journal/antioxidants/instructions

Examples:

  1. Page 1, line 28 “…processes (6).” Reference is not in square brackets [ ].
  2. From page 7 till 9, seems that square brackets of all references are in italic.
  3. The reference list is not in accordance with the Author’s Instructions. Example reference [1] P. Moi, K. Chan, I. Asunis, A. Cao, Y.W. Kan, Isolation of NF-E2-related factor 2 (Nrf2), a NF-E2-like basic leucine zipper transcriptional activator that binds to the tandem NF-E2/AP1 repeat of the beta-globin locus control region, Proc Natl Acad Sci U S A, 91 (1994) 9926-9930.

should be:

1. Moi, P.; Chan, K.; Asunis, I.; Cao, A.; Kan, Y. W. Isolation of NF-E2-related factor 2 (Nrf2), a NF-E2-like basic leucine zipper transcriptional activator that binds to the tandem NF-E2/AP1 repeat of the beta-globin locus control region. Proc. Natl. Acad. Sci. U. S. A. 1994, 91, 9926–30, doi:10.1073/pnas.91.21.9926.

Author Response

Dear reviewer 2,

We have corrected the reference-issued that you pointed out.

Reviewer 2:

The authors improved their manuscript. Yet, references are still not prepared under the journal’s Author Instructions:

https://www.mdpi.com/journal/antioxidants/instructions

Examples:

  1. Page 1, line 28 “…processes (6).” Reference is not in square brackets [ ].
  2. From page 7 till 9, seems that square brackets of all references are in italic.
  3. The reference list is not in accordance with the Author’s Instructions. Example reference [1] P. Moi, K. Chan, I. Asunis, A. Cao, Y.W. Kan, Isolation of NF-E2-related factor 2 (Nrf2), a NF-E2-like basic leucine zipper transcriptional activator that binds to the tandem NF-E2/AP1 repeat of the beta-globin locus control region, Proc Natl Acad Sci U S A, 91 (1994) 9926-9930.

should be:

  1. Moi, P.; Chan, K.; Asunis, I.; Cao, A.; Kan, Y. W. Isolation of NF-E2-related factor 2 (Nrf2), a NF-E2-like basic leucine zipper transcriptional activator that binds to the tandem NF-E2/AP1 repeat of the beta-globin locus control region. Proc. Natl. Acad. Sci. U. S. A.199491, 9926–30, doi:10.1073/pnas.91.21.9926.